# A Meta-Analysis on Quantitative Sodium, Potassium and Chloride Metabolism in Horses and Ponies

**DOI:** 10.3390/ani15020191

**Published:** 2025-01-13

**Authors:** Isabelle Maier, Ellen Kienzle

**Affiliations:** Animal Nutrition and Dietetics, Department of Veterinary Sciences, Ludwig-Maximilians-Universität München, Schoenleutnerstr. 8, D-85764 Oberschleissheim, Germany; kienzle@tiph.vetmed.uni-muenchen.de

**Keywords:** pony, horse, mineral, metabolism, sodium, potassium, chloride, digestibility

## Abstract

The current meta-analysis reviewed the literature researching the metabolism of sodium (Na), potassium (K) and chloride (Cl) regarding possible differences between ponies and horses, the effect of exercise and the impact of water deficiency. It was found that working equines were able to digest Na and, to a lesser extent, K more effectively than non-working animals. Regarding Cl availability, there was no difference between working equines and those in maintenance. Ponies were able to digest Cl better than horses, but without any relevance for practical feeding. Water deficiency affected the K and Cl metabolism to a minor extent, while the Na metabolism remained unaffected.

## 1. Introduction

Na, K and Cl are crucial minerals in metabolism. Na and K are responsible for the generation of cell membrane potentials and, in this context, are essential for muscle and nerve cell functionality [1]. Additionally, Na plays a key role in blood pressure regulation [2].

Na deficiency in horses may lead to behavioural changes, such as excessive licking, affect the central nervous system, and can cause seizures [3,4]. Excessive Na and Cl intake due to intense salt supplementation can be associated with gastric ulcers [5], fluid retention in tissues and metabolic acidosis [6]. Muscle weakness and paralysis can be caused by K deficiency [7]. However, a K deficiency is almost non-existent in equines [8], as rations for horses are typically rich in K [9]. One factor that could contribute to K deficiency is soaking hay, as this process can lead to significant K losses [10,11,12]. A considerable excess of K can lead to disturbances in heart function [13] and can be involved in the development of hyperkalemic periodic paralysis (HYPP) in predisposed horses [14]. Cl deficiency has been documented to lead to metabolic alkalosis [8]. However, it is likely that this is not primarily caused by Cl deficiency itself, but rather by the opposite changes in bicarbonate concentration that typically accompany alterations in Cl balance [14]. For example, in hard-working horses experiencing excessive chloride losses, it has been reported that bicarbonate is reabsorbed by the kidneys to maintain electrical neutrality and that this bicarbonate reabsorption can lead to the development of metabolic alkalosis [15].

The recommendations for the electrolyte supply of horses show an enormous range. For instance, the recommended Na requirements for maintenance for a 500 kg horse range from 3 to 13 g per day [16,17,18]. For working horses, the differences are even more relevant. The recommended intake for a middle-sized horse engaged in hard work ranges from 22 to about 113 g Na per day [16,17,18,19]. Revisiting the underlying data might help explain the different recommendations.

Faecal endogenous losses, apparent and true digestibility of minerals as well as data on renal excretion and retention are important factors for factorial calculation of mineral requirements for maintenance [18,20]. A previous meta-analysis by Kienzle and Burger [20] showed a low true digestibility of Na in horses. The same was observed for other large hindgut fermenters [21,22,23]. In the present meta-analysis, new data were included. In light of a previous meta-analysis by Maier and Kienzle [24], which identified significant differences between ponies and horses regarding apparent Mg digestibility, the present meta-analysis aimed to evaluate whether such differences also exist for the minerals Na, K and Cl. In addition to apparent digestibility, particular attention was given to potential disparities in true digestibility and renal excretion. Furthermore, the effect of exercise on the aforementioned parameters was also investigated. To the best of our knowledge, no data are currently available on the extent to which physical activity influences the digestibility of Na, K and Cl. A meta-analysis is particularly well suited for this purpose, making it a valuable aspect to address within the scope of the present study. In two studies, experiments were conducted under water restrictions. Since electrolytes, particularly Na, are closely interconnected with water balance [2], these data were specifically marked, and the impact of water restrictions on the mentioned parameters was also analysed. Coincidentally, data were found from three studies with ponies which were later diagnosed with PPID. This enabled a comparison of the quantitative metabolism of the three minerals mentioned in preclinical PPID with other data.

## 2. Materials and Methods

The following databases were used for literature research: Google Scholar, the database information system (DBIS) of the LMU Munich, PubMed and the Online Public Access Catalogue (OPAC) of the LMU Munich and the Bavarian State Library (BSB). “Mineral digestibility, sodium, potassium, chloride, horse, pony, mare, gelding, salt, sodium chloride, saline, salted, digestibility, availability, minerals, set elements, electrolytes, renal, kidney, faecal, serum, blood, excretion, resorption or absorption” were the main keywords searched for in different constellations. Studies that were used in a previous meta-analysis were also included [20]. Only data from studies including mature equines (aged over 36 months, as defined by the Society of Nutrition Physiology (GfE) [18]), which were not lactating or pregnant, were considered. The analysis involved both equines at maintenance and those during work (if the animals were exercised, as stated in the respective study), which was indicated accordingly. Experiments with water deficiency (one trial by Hipp-Quarton [25] and four trials by Pérez-Noriega [26]) were also considered and marked separately.

Analogous to the study by Maier and Kienzle [24], only studies that provided data on at least two of the following parameters were included: daily mineral intake, apparent digestibility, faecal excretion, renal excretion, and serum blood levels. Information on the body weight or at least the breed of the animals had to be disclosed to estimate the body weight. This estimation was based on the body mass in kg of large adult horses with a body condition score (BCS) of 5–6 [18]. Additionally, details on age, breed and feed composition were required. Both individual data and group averages were considered.

This meta-analysis followed the guidelines of the Systematic Reviews and Meta-Analyses (PRISMA) statement [27]. The most recent source search was completed in September 2024. The following studies were utilized: [3,25,26,28,29,30,31,32,33,34,35,36,37,38,39,40,41,42,43,44,45,46,47,48,49,50,51,52,53,54,55,56,57]. A flowchart illustrating the search and selection process is provided in Appendix A, along with Appendix A, which details the number of studies and participants used in each graph.

The division into pony and horse was based on body weight (ponies < 300 kg; horses > 300 kg).

The requirement recommendations set by GfE [18] were applied. The reference range for serum mineral concentration was taken from the book *Equine Applied and Clinical Nutrition: Health, Welfare and Performance* [58].

For the meta-analysis, all data on intake and excretion were calculated per kilogram (kg) of metabolic body weight (MBW). Plots were created similar to those in the study by Maier and Kienzle [24]. For all minerals, the Lucas test [20,59,60], with which an apparently digested amount of a nutrient is plotted against intake, was used. Given that the data distribution is appropriate for linear regression analysis, multiplying the regression coefficient by 100 will indicate true digestibility (in%), while the intercept will indicate the endogenous losses. The mineral requirement to replace faecal losses is then calculated by dividing the endogenous losses by the true digestibility and multiplying by 100 [24].

Data on apparent mineral digestibility, serum concentration, renal excretion, and retention were plotted against the intake of the respective mineral. The retention included electrolyte losses via sweat in working animals and was calculated as follows: mineral intake minus faecal losses minus urinary losses. An evaluation of retention, excluding electrolyte losses through sweat (calculated as mineral intake minus faecal losses, urinary losses and sweat losses), was also conducted for those studies where data on sweat losses were available.

Outliers, defined by three standard deviations, appeared in all serum plots and were therefore eliminated from these diagrams. One of these outliers concerned the data from Baker [28], whose serum levels for all minerals were in a range that could have caused clinical issues. It is likely that there is an error in the units at some point. Additionally, the data of Neustädter for Na serum levels [43] and the data of O’Connor for the K serum level plot [44] produced outliers. Both used different measurement methods than the other studies and were therefore not considered for the aforementioned graphs.

In almost all Cl diagrams, the data of Schryver [49] represented outliers. In addition, the Cl balance given in this publication was positive, but renal excretion was higher than intake. Presumably, there is some error in these data; therefore, the data were removed from all Cl plots.

In the studies of Stürmer [52], Berchtold [31] and Schiele [47], two ponies were included, which were, after the completion of the study, diagnosed with PPID. An examination was conducted to determine whether any abnormalities existed in Na digestibility for these ponies. No significant differences were found. Therefore, the aforementioned studies were used for the calculations of Na. For K, the affected animals showed an exceptionally low K digestibility of around 60%, even before diagnosis. In the study of Schiele [47], one animal consuming overhanging hay had an apparent K digestibility of only 44%. In the other studies stated above, only the average across all four ponies used was provided. Serum K levels were within the reference range. Renal excretion in relation to intake but not to the apparently digested amount of K showed a difference between the affected and unaffected ponies. The mean retention, however, was clearly negative (−215 ± 16 mg/kg MBW) compared to all other studies (55.9 ± 34.2 mg/kg MBW). Therefore, the studies of Stürmer [52], Berchtold [31] and Schiele [47] were removed from the calculations on K. Data on Cl, including the affected ponies (i.e., Stürmer [52] and Berchthold [31]), were also checked for systematic differences. The apparent digestibility of Cl was significantly higher than in other studies (*p* = 0.03). There were no data on serum Cl. Renal excretion as a percentage of intake for the studies stated above was significantly lower compared to all other studies (*p* = 0.03), while the retention plus sweat losses as a percentage of intake was significantly higher (*p* = 0.01). In turn, these studies were not considered for investigations regarding Cl.

The comparison of two means was performed using Student’s *t*-test or, if the data were not normally distributed, the Mann–Whitney rank sum test was applied. For analyses involving more than one contributing factor, a two-way ANOVA was conducted utilizing the Holm–Sidak post hoc test for all pairwise comparisons. The aforementioned tests were conducted using SigmaPlot 14 (Systat Software, San Jose, CA, USA). Hyperbolic regressions were computed as nonlinear inverse first-order regressions using SigmaPlot. Linear regressions were also determined using SigmaPlot. Comparisons of linear regression lines were made using the BiAS program with the test of Ho (BiAS. für Windows, Version 11.01, 2023, epsilon-Verlag, Frankfurt, Germany). The significance level was defined at <0.05.

## 3. Results

### 3.1. Sodium

#### 3.1.1. Na Digestibility

In Figure 1, the apparent digestibility of Na is plotted against Na intake. It shows a typical hyperbolic curve with low apparent digestibility at an intake around or below the requirement recommendations (27 mg/kg MBW, following GfE [18]). Working ponies and horses appeared to have a higher apparent Na digestibility, especially at a low range of intake. For a medium level of work (30% above maintenance), the recommended Na requirement amounts to approximately 120 mg/kg MBW [18]. In the range below a Na intake of 120 mg/kg MBW, there was a highly significant difference in apparent Na digestibility between working and non-working horses (median working: 65%, n: 39; median not working: 33%, n: 135; *p* < 0.001). There was no significant difference in apparent Na digestibility between ponies and horses across the whole range of intake (median pony: 58%, n: 198; median horse: 52%, n: 164; *p* = 0.08). The same was true if only trials with an intake above the requirements were considered.

The amount of apparently digested Na in relation to intake is shown in Figure 2. Dark green dots mark the animals that were exercised during the experiments. Non-working ponies and horses are represented by light green triangles. Trendlines were calculated for either working or non-working ponies and horses (Figure 2). There was a significant difference between these trendlines (*p* = 0.003), which represent true digestibility when multiplied by 100. The standard error of the mean (SEM) of the regression line was much higher in the non-working animals.

#### 3.1.2. Serum Na Concentration

Figure 3 shows the serum Na concentration in relation to Na intake. The serum Na level was almost the same, both above and below the recommended requirements of 27 mg/kg MBW following GfE [18] (mean up to this requirement = 134.88 mg/kg MBW; mean above this requirement = 135.98 mg/kg MBW). There was no effect of being a pony or a horse or of working or not working. The distribution of the data did not allow for a statistical evaluation using ANOVA. The data on animals with water deficiency did not deviate from the overall dataset.

#### 3.1.3. Na Renal Excretion

There was no effect of being a pony or horse on renal Na excretion in relation to intake or to apparently digested Na. Working or not working, however, displayed a clearcut effect in both cases. Figure 4 shows the renal Na excretion in relation to apparently digested Na. It was visibly higher in non-working ponies and horses.

#### 3.1.4. Na Retention (Plus Sweat Losses)

Na intake minus faecal and renal excretion represents retention and sweat losses. In Figure 5, this parameter is plotted against Na intake. Retention in non-working ponies and horses rose with increasing intake. At very high intakes, however, it decreased again. This happened in ponies and in horses; however, the effect was stronger in horses. A two-factorial ANOVA of Na retention plus sweat losses in percentage of Na intake with the factors pony vs. horse and working vs. not working showed a significant difference between working and non-working ponies and horses (*p* < 0.001). Working horses had a higher retention plus sweat losses in percentage of intake than working ponies (horse working: mean = 24%; pony working: mean = 15%). In non-working animals, it was the opposite: horses had a more negative retention in percentage of intake than ponies (horse not working: mean = −22%; pony not working: mean = −16%). The retention in non-working animals in percentage of intake averaged −18%. The retention plus sweat losses in working animals in percentage of intake averaged 17%.

After accounting for sweat losses, the retention of Na was predominantly negative, whereas the losses through sweat were comparatively high (Figure 6). The apparent digestibility in the studies considered averaged 76%.

### 3.2. Potassium

#### 3.2.1. K Digestibility

In Figure 7, the apparent K digestibility is plotted against K intake. A typical hyperbolic curve emerged. Data from trials with a lack of water did not exhibit any anomalies. The regression lines between ponies and horses did not show any difference.

Figure 8 shows the apparently digested K quantity in relation to K intake. There was an almost linear relationship between the factors. Studies with water deficiency did not produce outliers in this plot. Trendlines were calculated for either working or non-working ponies and horses (Figure 8). There was a significant difference between these trendlines (*p* < 0.001). The same applied when the data were considered up to an intake of 2500 mg/kg MBW. Above this intake level, only working animals were represented due to the data distribution. No differences between ponies and horses were observed for the non-working animals (*p* = 0.47). Similarly, there was no significant effect of being a pony or horse among the working animals (*p* = 0.48).

#### 3.2.2. Serum K Concentration

Figure 9 shows the serum K levels in relation to K intake. At an intake below the recommended requirements (139 mg/kg MBW [18]), K serum was partly below the minimum reference range; otherwise, the K intake showed no influence. There was no effect of being a pony or a horse (*p* = 0.34).

#### 3.2.3. K Renal Excretion

The renal K excretion shows an almost linear increase with rising intake (Figure 10). With an intake of around 0 mg/kg MBW, renal K excretion was almost non-existent. The recommended K requirement for maintenance is 139 mg/kg MBW [18]. At an intake of more than 2500 mg/kg MBW, only data on ponies were available due to the data distribution. A two-factorial ANOVA of renal K excretion with the factors pony vs. horse and working vs. not working at an intake of 139–2500 mg/kg MBW showed no difference between ponies and horses or working and non-working ponies and horses. There was no significant interaction between these factors (*p* = 0.26). Data from studies with water deficiency showed renal excretions in the upper range.

#### 3.2.4. K Retention Plus Sweat Losses

In Figure 11, retention of K (including sweat losses) is plotted against K intake. Retention plus sweat losses in working and non-working ponies and horses increased with increasing intake. At very high intakes, however, it decreased again in a pattern similar to that of Na. At an intake range of 139–2000 mg/kg MBW, horses showed significantly higher retention and sweat losses than ponies. There was a significant difference between working and non-working ponies and horses (two-factorial ANOVA of K retention plus sweat losses at an intake range of 139–2000 mg/kg MBW with the factors pony vs. horse and working vs. not working: mean ponies = 65.3 mg/kg MBW; mean horses = 150.9 mg/kg; mean working = 157 mg/kg MBW; mean not working = 59.3 mg/kg MBW; *p* = 0.02). With a K intake of above 2000 mg/kg MBW, the data distribution did not allow for a statistical evaluation for the amount of K retention plus sweat losses or for the factors pony or horse and working or not working. Data from experiments with water deficiency were significantly lower compared to other data in the aforementioned range (*p* = 0.001; Figure 11). So were the results of Baker [28], who added K as K salts, whereas in all other experiments, K originated from plants.

K sweat losses and K retention after deducting sweat losses showed a pattern similar to that of Na, yet the K retention remained entirely in the negative range (Figure 12).

### 3.3. Chloride

#### 3.3.1. Cl Digestibility

In Figure 13, the apparent digestibility of Cl is plotted against Cl intake. At an intake of more than 250 mg/kg MBW, the apparent digestibility was greater in ponies than in horses. The difference was significant (median pony = 98%, median horse = 94%, *p* < 0.001).

At low intake rates, the amount of apparently digested Cl plotted against Cl intake (Figure 14) showed an almost linear relationship between these factors. At an intake of above 600 mg/kg MBW, the apparently digested Cl quantity seemed to be higher in ponies than in horses. Trendlines were calculated separately for ponies and horses. There was a significant difference between these trendlines (*p* < 0.001). The factors working and not working did not show any significant differences (*p* = 0.18).

#### 3.3.2. Serum Cl Concentration

The Cl serum concentration in relation to intake is shown in Figure 15. There was no effect of being a pony or a horse. The serum Cl levels of experiments with water deficiency were above the reference range (Figure 15).

#### 3.3.3. Cl Renal Excretion

In Figure 16, the renal excretion of Cl is plotted against Cl intake. The data show a broad distribution. Working ponies and horses seemed to have a lower renal excretion than non-exercised animals. A two-way ANOVA of renal Cl excretion as a percentage of intake with the factors pony vs. horse and working vs. not working showed a significant difference between working and non-working ponies and horses (mean working = 61 ± 4%, mean not working = 93 ± 4%, *p* = 0.02).

#### 3.3.4. Cl Retention Plus Sweat Losses

Figure 17 shows the retention and sweat losses of Cl in relation to Cl intake. Retention plus sweat losses increased with increasing intake. The retention of working ponies and horses was significantly higher than the retention of non-working horses and ponies (mean working: 217.8 ± 38.4 mg/kg MBW; mean not working: −12.8 ± 38.9 mg/kg MBW; *p* < 0.001). In non-working ponies and horses, the retention in percentage of intake averaged 3%. The retention plus sweat losses in working ponies and horses in percentage of intake averaged 26%.

The Cl retention shown in Figure 18 appears extremely negative, accompanied by very high sweat losses.

## 4. Discussion

As already described by Maier and Kienzle [24], apparent digestibility defines the proportion of a consumed nutrient that is not excreted via the faeces. It does not consider endogenous faecal losses. If these endogenous faecal losses are higher than intake, the apparent digestibility can be negative, despite the absorption of a substantial amount of the nutrient from the feed (see Figure 1, Figure 7 and Figure 13). In addition to measuring apparent digestibility, it is also possible to plot the apparently digested amount of a nutrient against intake. This method, known as the Lucas test, has been used multiple times in the past [20,24,60,61]. Serum concentration and renal excretion, as well as retention plus sweat losses, can also be shown in relation to intake, allowing for the identification of quantitative similarities and differences. In the present meta-analysis, these methods were employed.

The distinction between pony and horse using a cut-off of 300 kg is a rather unrefined method. As described in a previous meta-analysis by Maier and Kienzle [24], it is possible that some heavier ponies are classified as horses. Nevertheless, there is no alternative method in a meta-analysis without significantly limiting the data pool.

The same is true for the differentiation between working and non-working equines. Work was not graded into light, medium or hard work. Estimating sweat losses based on work intensity is not possible, as actual losses are heavily influenced by environmental factors such as temperature or humidity [62]. For the same reason, it is equally impossible to infer the intensity of work based on sweat losses. Therefore, retention was analysed as retention plus sweat losses. However, data on electrolyte losses through sweat were available for five studies involving working ponies [25,26,33,36,38]. Consequently, these studies were used in a separate analysis to evaluate retention after deducting sweat losses.

Differences between ponies and horses have been investigated in the past, for example, with regard to the microbiome [63,64,65] or the digestibility of crude nutrients [66,67,68,69]. However, to the best of our knowledge, the digestibility of minerals has not yet been investigated in relation to differences between ponies and horses, except for an analysis by Maier and Kienzle [24], or in relation to the association between digestibility and work. While there is a wide range of studies on the influence of exercise and electrolyte losses through sweat in general [62,70,71,72,73,74,75,76,77], none of these studies have measured digestibility.

Presenting all data per kg of MBW has been discussed extensively for species with highly variable adult body weights, such as horses and dogs [18,78,79,80]. To ensure accuracy, all diagrams were recreated using the reference measure of kg body weight (BW). No differences were observed.

There was no difference between ponies and horses in terms of apparent Na digestibility or apparently digested Na (Figure 1) in relation to intake. Both parameters, however, differed between working and non-working animals, with working animals having a higher apparent digestibility. This also applied to true Na digestibility. Besides, the endogenous losses were higher in animals that were not exercised. Especially in non-working animals, the apparent and true digestibility of Na was relatively low compared to data from other species such as dogs, cats, pigs and mice [81,82,83,84,85]. In horses and other large hindgut fermenters (Asian elephants, black rhinoceros), a rather low Na digestibility was described by Clauss et al. [21,86]. Some of the black rhinoceros in the study by Clauss et al. [21] had extremely low Na digestibility, even though the Na content in their diet was not very low. Also, tapirs seem to be very similar in Na digestibility to horses [22]. In a study by Holdø et al. [23], African elephants showed higher Na excretion in faeces in Na-rich areas compared to lower faecal Na excretion in elephants from areas with comparatively less Na. These findings suggest that several hindgut fermenters, including horses, may be somewhat different from other species with regard to regulation of Na metabolism. The suggested hypothesis is that the down- and upregulation of Na absorption from the gastrointestinal tract plays a bigger role in horses than in species such as dogs, humans and mice, where Na metabolism is not that strongly regulated via the intestine but much more through renal mechanisms [87,88,89]. As shown in Figure 4, this does not imply that the kidney is not involved in regulation in equines at all. Rather, it is likely involved to a lesser extent, while the gastrointestinal tract also seems to play a significant role. The theory of increased regulation of Na absorption in the case of an increased Na requirement (i.e., work) is also supported by the lower variation of the data in working animals compared to non-working horses and ponies (SEM trendline working = 9.523; SEM trendline not working = 18.378).

A difference in Na digestibility between working equines and animals at maintenance has a strong effect on the factorial calculation of requirements, especially for working animals. In factorial calculation, the endogenous losses and sweat losses are summed up and divided by true digestibility. If the true digestibility is higher, this will result in lower figures. This considerable up and downregulation of intestinal Na absorption in horses may in part explain why working horses appear to remain healthy, even if their factorially calculated Na requirements are not completely met [90].

This may also explain why electrolyte loading for several days before strenuous work is not recommended [91]. Presumably, it leads to downregulation of intestinal Na absorption. The findings of Zeyner et al. [6] also strengthen this hypothesis. They fed either 0, 50 or 100 g (g) of salt (NaCl) per day to exercising horses and found stronger effects with a lower dose. This finding could in part be explained by differences in intestinal absorption, with lower absorption at higher doses. It also explains the findings on Na retention plus sweat losses (Figure 5) in experiments with very high Na intake.

Renal Na excretion in comparison to intake and Na retention in working or non-working animals reflects sweat losses. The differences between working ponies and horses suggest lower sweat losses in ponies than in horses. While the sweat composition is not systematically different, horses appear to produce more sweat when adrenaline is injected [92]. Therefore, the hypothesis that horses produce more sweat during work appears to be a likely explanation. Na retention excluding quantitative Na losses via sweat could only be investigated for studies with ponies where quantitative data on sweat losses were available. Due to the lack of data on sweat losses in working horses, it is unfortunately not possible to assess the difference between ponies and horses in this context for all minerals. Nonetheless, the very low Na retention combined with a relatively low apparent Na digestibility is an interesting finding. In comparison to other mammals such as dogs, where the apparent Na digestibility can reach up to 95% [81], horses do not appear to achieve this even with significant sweat losses and low retention. The capacity for Na absorption through the intestine seems to be limited in horses, even in working animals. The same is true for the reduction of renal losses.

Apparent and true K digestibility in horses and ponies was higher than in other large hindgut fermenters such as black rhinoceros and tapirs [21,93]. It was closer to other species such as pigs, dogs, cats and mice [81,83,85,94]. Work increased the apparent and true K digestibility of horses and ponies, presumably because of a higher requirement for sweat losses, which resulted in an upregulation of intestinal absorption. Compared to Na, the effect was smaller, but it should still be considered for factorial calculation of requirements. In contrast to Na, in K, the status of deficiency is reflected by serum K, which affirms the information provided by Vervuert and Kienzle [95]. The same is already known from other species such as dogs, cats and humans [96,97]. K sweat losses plus retention were higher in working horses than in working ponies, suggesting higher sweat losses in horses. The explanation given for the differences in Na is probably also applicable to K.

The low K retention plus sweat losses observed during water deficiency could be the result of activation of the renin–angiotensin–aldosterone system because thirst leads to the release and activation of various hormones, ultimately resulting in the production of aldosterone [98]. This hormone increases the reabsorption of Na into the blood through Na-K pumps, which simultaneously leads to an increased excretion of K [99]. Similar to Na, K retention excluding K sweat losses was negative, with relatively high losses through sweat. The interpretation of the results is in accordance with that of Na.

The apparent and true digestibility of Cl were extremely high. This is in agreement with studies on other species such as dogs and cats [81,94]. There was a difference between horses and ponies. Given the overall extremely high digestibility, the physiological significance of this difference is unlikely to be of practical importance. An explanation might be the allometric relationship between the size of the digestive tract and the MBW, which is larger in ponies than in horses. The empty gastrointestinal tract amounts to 5% of BW [8]. With increasing size, the surface of the gastrointestinal tract then decreases in relation to MBW, provided there is no compensation by an increase of villi and other structures which enlarge the resorptive surface. To the best of our knowledge, this is not the case.

Serum Cl levels did not reflect the intake, as repeatedly reported before [95]. While the same was true for Na, there was a difference between Na and Cl during water deficiency. The serum Na concentration did not increase, but Cl serum levels did. This suggests a less strict regulation of Cl metabolism compared to Na.

Renal excretion of Cl clearly showed effects on intake. Working animals had lower renal excretion in relation to intake, suggesting sweat losses. Cl retention and sweat losses reflected the other Cl results. The data distribution did not allow for a closer investigation of Cl retention with respect to the potential difference between ponies and horses. To the best of our knowledge, any such potential difference has not yet been investigated in the literature. The sweat losses of Cl were extremely high (Figure 18). This is consistent with previous reports indicating that Cl is lost in quantitatively high amounts through sweat [15,77]. As a result, Cl exhibits the most negative retention compared to Na and K. But the Cl digestibility was 93% in the studies considered, with renal Cl losses also being very high. This provides interesting material for further investigations.

As described in the Section 2, some studies included animals later diagnosed with PPID. With regard to clinical use of the data, it is worth discussing the findings in these studies in comparison to all other studies. The most important difference was a very low apparent K digestibility, of around 60% and less, compared to above 80% in the other studies. In factorial calculation, this results in a requirement which is around 30% higher than in healthy animals. In horse feed, especially in forage, K is normally abundant and a deficiency is therefore virtually impossible [100]. Horses with PPID are mostly older horses [101,102], which may be fed soaked hay to remove soluble carbohydrates and or to remove dust from their diet. Soaking the hay may remove more than 50% of the K content [10,11,12]. Therefore, in horses with PPID eating soaked hay, the resulting K intake may become marginal. In contrast to K, Na metabolism in horses with PPID appeared to be completely unaffected, whereas in other species, Cushing patients are reported to have increased Na retention [103,104,105,106]. In the studies with elderly horses, Cl retention increased, a finding which is not an eminent feature of Cushing’s disease in other species.

The observed increased digestibility of Na and K in working equines, combined with the still highly negative retention, provides an important basis for further research.

## 5. Conclusions

The apparent Na digestibility in horses and ponies was relatively low, similar to other hindgut fermenters. During physical activity, both apparent and true Na digestibility increased considerably, suggesting a more important role of the gut in the regulation of Na metabolism in equines than in other species. A similar effect was observed to a lesser extent for K digestibility. For Cl, both apparent and true digestibility were notably high, with ponies exhibiting even greater digestibility than horses. However, given the overall high Cl digestibility in both groups, this finding holds little relevance for practical feeding. Water deprivation led to increased serum Cl levels and decreased K retention due to relatively elevated renal K excretion, likely influenced by aldosterone. In the ponies later diagnosed with PPID, no impact on Na metabolism was detected. This contrasts with the findings observed in other species. Furthermore, reduced K digestibility was noted in PPID-affected animals, but there were no changes in blood K levels. Cl retention increased in animals later diagnosed with PPID, which has not been previously reported in other species.

## Figures and Tables

**Figure 1 animals-15-00191-f001:**
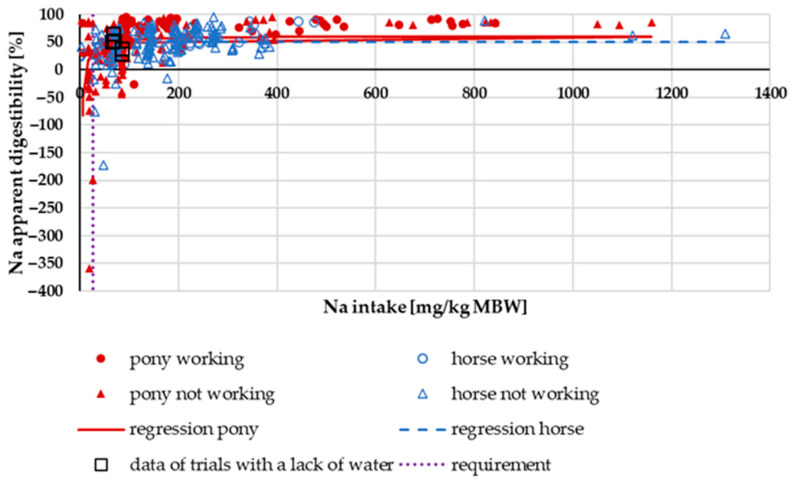
Relationship between Na intake (in mg/kg MBW) and apparent Na digestibility (in%). Regression lines mark the data of horses compared to ponies.

**Figure 2 animals-15-00191-f002:**
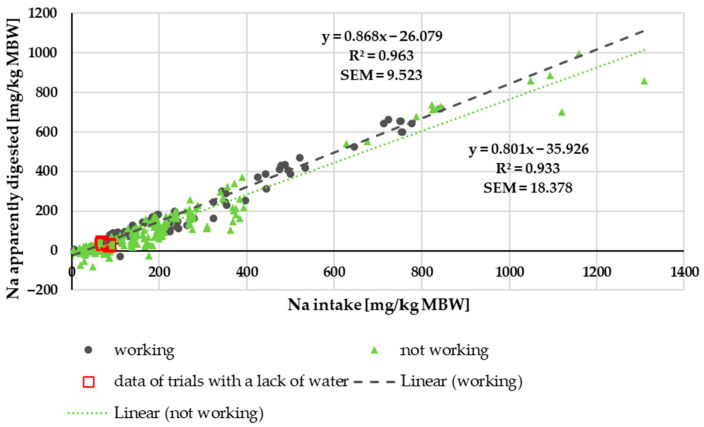
Apparently digested Na in relation to Na intake (both in mg/kg MBW). Trendlines mark working and non-working horses and ponies.

**Figure 3 animals-15-00191-f003:**
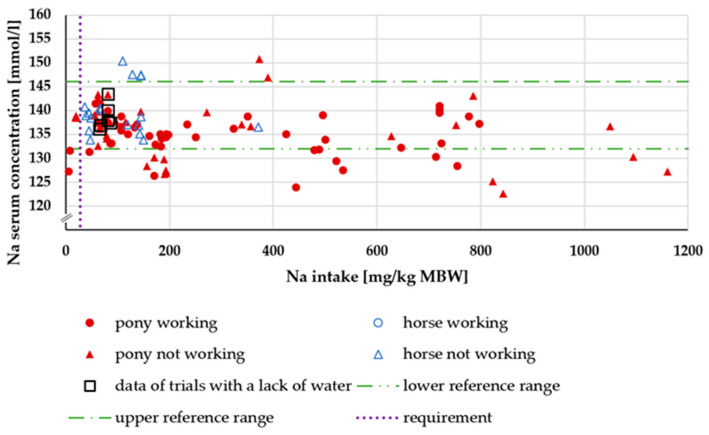
Serum Na concentration (in mmol/L) potted against Na intake (in mg/kg MBW).

**Figure 4 animals-15-00191-f004:**
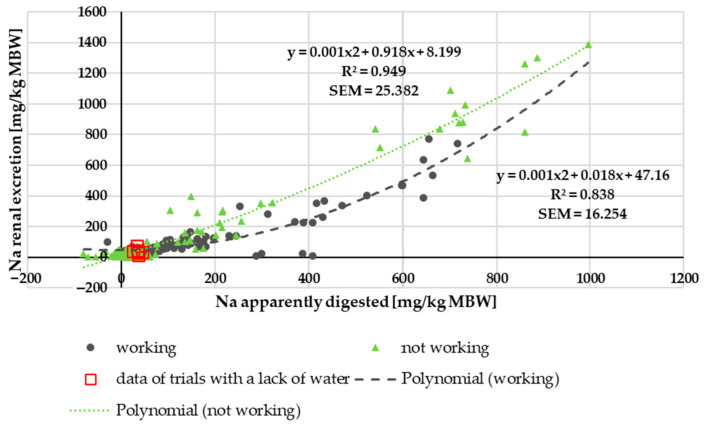
Relationship between apparently digested Na and renal Na excretion (both in mg/kg MBW). Trendlines mark working and non-working horses and ponies.

**Figure 5 animals-15-00191-f005:**
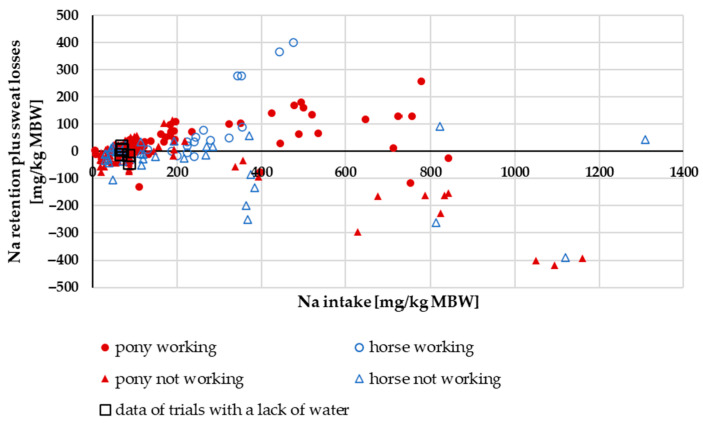
Relationship between Na intake and Na retention plus sweat losses (both in mg/kg MBW).

**Figure 6 animals-15-00191-f006:**
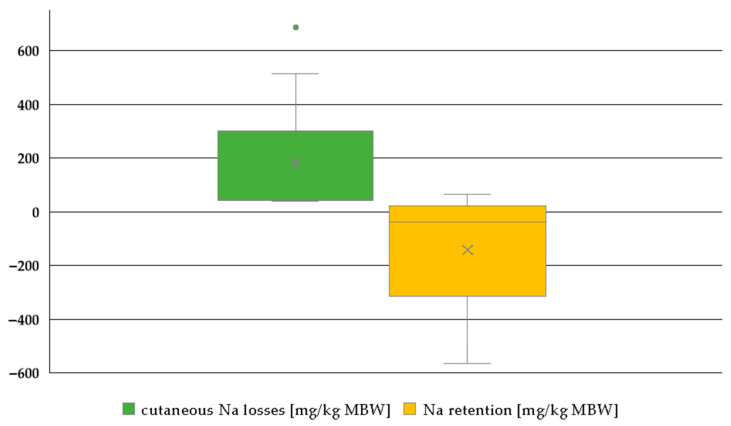
Box plot of Na sweat losses and Na retention (both in mg/kg MBW). The horizontal line represents the median, while the cross indicates the mean. Only data on working ponies, including information on Na sweat losses, were available [25,26,33,36,38]. Trials with water deficiency were excluded.

**Figure 7 animals-15-00191-f007:**
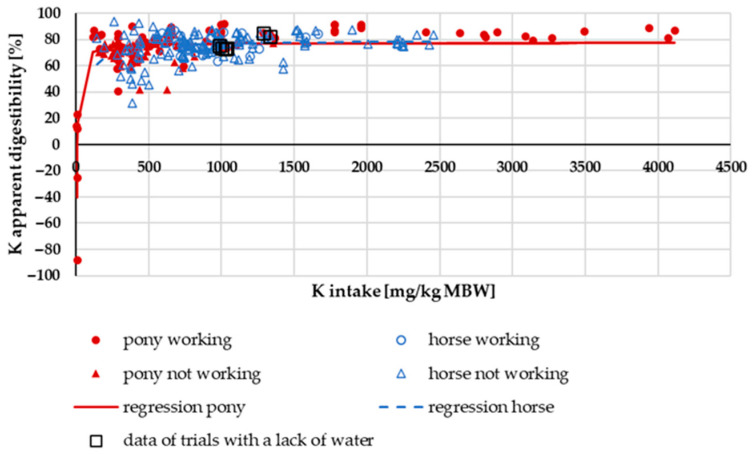
Apparent K digestibility (in%) plotted against K intake (in mg/kg MBW). The data of trials that included ponies later diagnosed with PPID were not considered. Regression lines mark the data of horses compared to ponies.

**Figure 8 animals-15-00191-f008:**
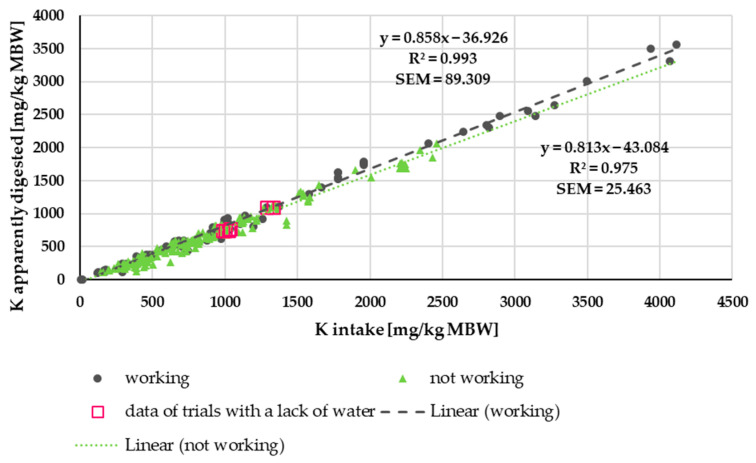
Relationship between K intake and apparently digested K (both in mg/kg MBW). The data of trials that included ponies later diagnosed with PPID were not considered. Trendlines mark working and non-working horses and ponies.

**Figure 9 animals-15-00191-f009:**
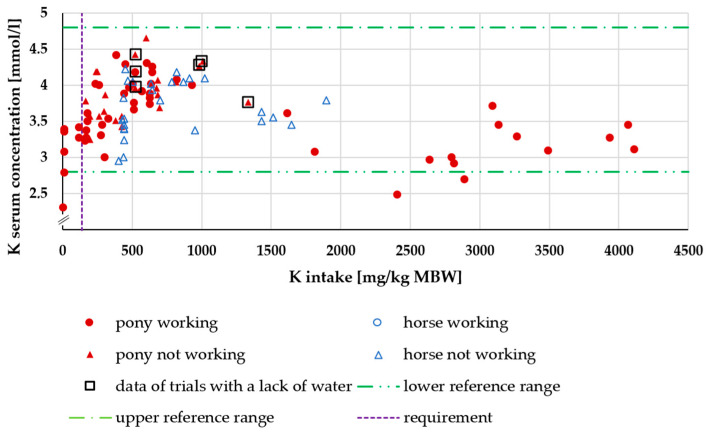
Serum K concentration (in mmol/l) in relation to K intake (in mg/kg MBW). The data of trials that included ponies later diagnosed with PPID were not considered.

**Figure 10 animals-15-00191-f010:**
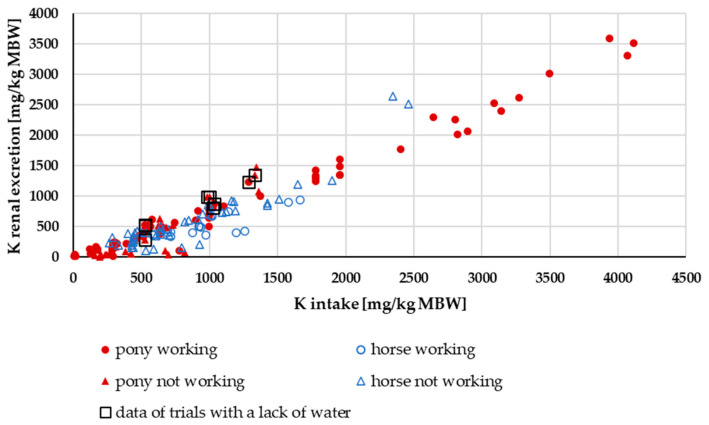
Renal K excretion plotted against K intake (both in mg/kg MBW). The data of trials that included ponies later diagnosed with PPID were not considered.

**Figure 11 animals-15-00191-f011:**
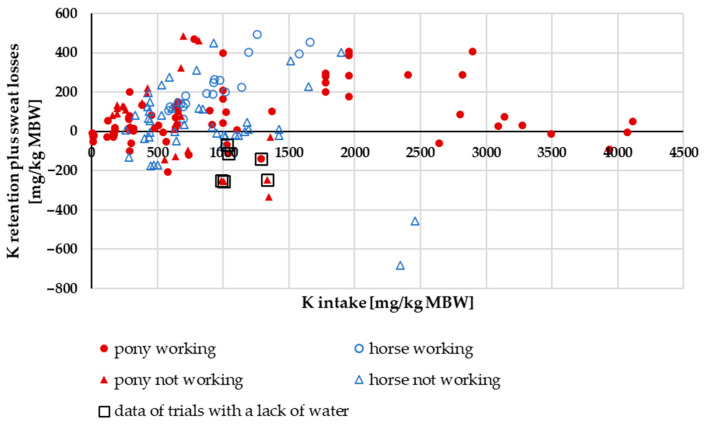
K retention plus sweat losses in relation to K intake (both in mg/kg MBW). The data of trials that included ponies later diagnosed with PPID were not considered.

**Figure 12 animals-15-00191-f012:**
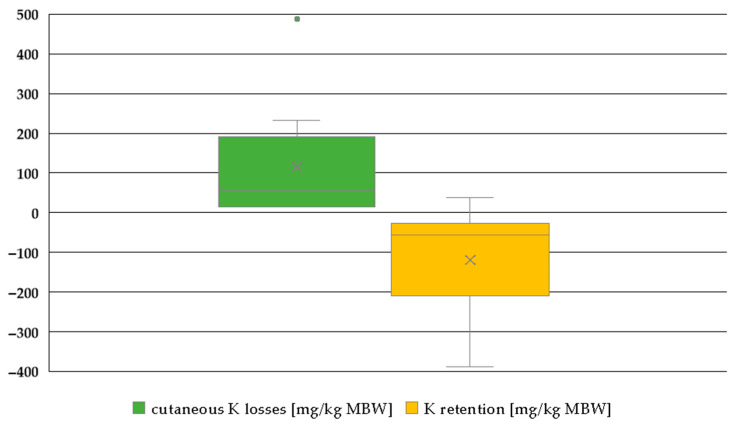
Box plot of K sweat losses and K retention (both in mg/kg MBW). The horizontal line represents the median, while the cross indicates the mean. Only data on working ponies, including information on K sweat losses, were available. Trials with water deficiency were excluded.

**Figure 13 animals-15-00191-f013:**
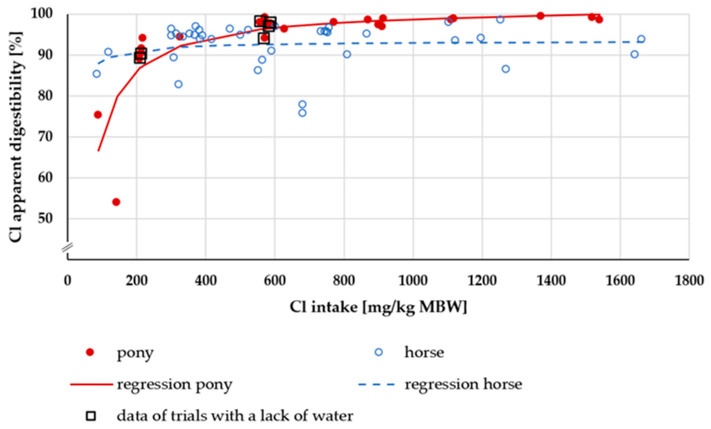
Relationship between Cl intake (in mg/kg MBW) and apparent Cl digestibility (in%). The data of trials that included ponies later diagnosed with PPID were not considered. Regression lines mark the data of horses compared to ponies.

**Figure 14 animals-15-00191-f014:**
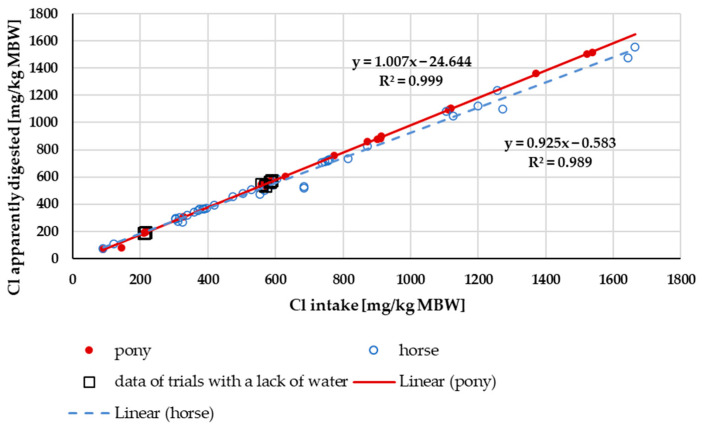
Relationship between Cl intake and apparently digested Cl (both in mg/kg MBW). The data of trials that included ponies later diagnosed with PPID were not considered. Trendlines mark the data of horses compared to ponies.

**Figure 15 animals-15-00191-f015:**
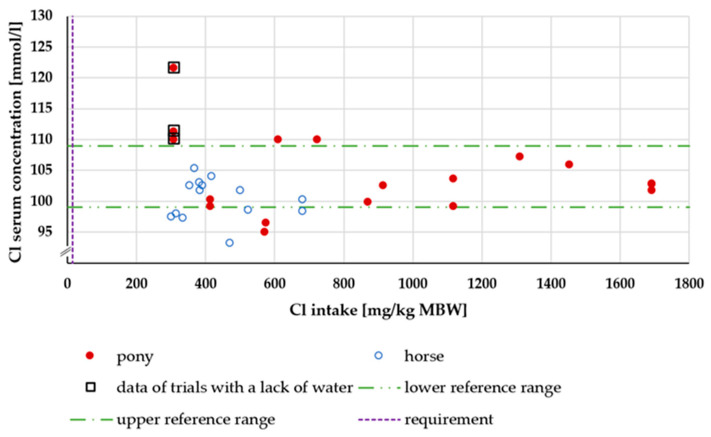
Serum Cl concentration (in mmol/l) plotted against Cl intake (in mg/kg MBW).

**Figure 16 animals-15-00191-f016:**
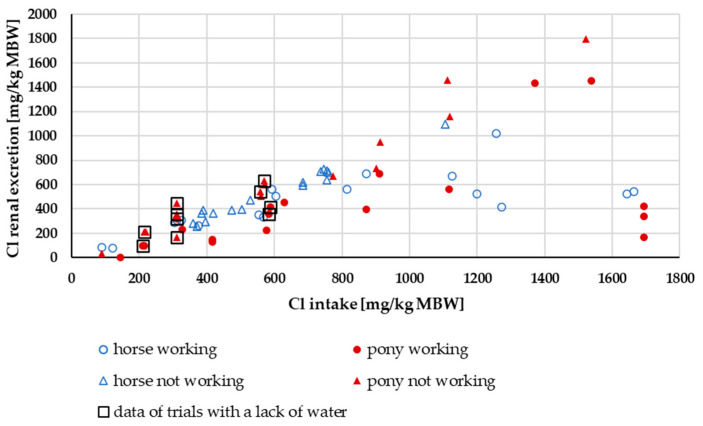
Renal Cl excretion in relation to Cl intake (both in mg/kg MBW). The data of trials that included ponies later diagnosed with PPID were not considered.

**Figure 17 animals-15-00191-f017:**
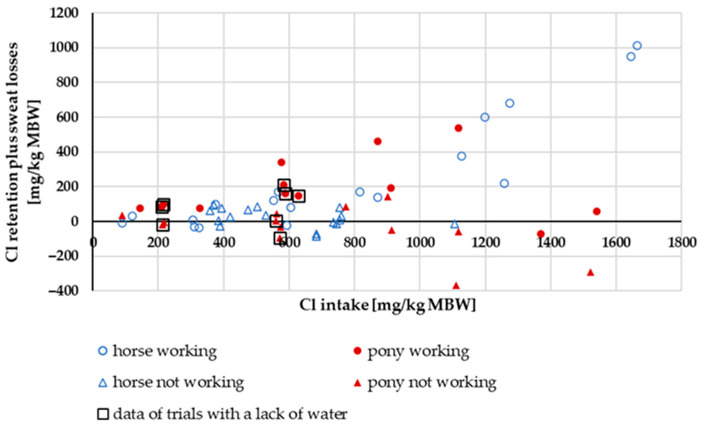
Cl retention plus sweat losses in relation to Cl intake (both in mg/kg MBW). The data of trials that included ponies later diagnosed with PPID were not considered.

**Figure 18 animals-15-00191-f018:**
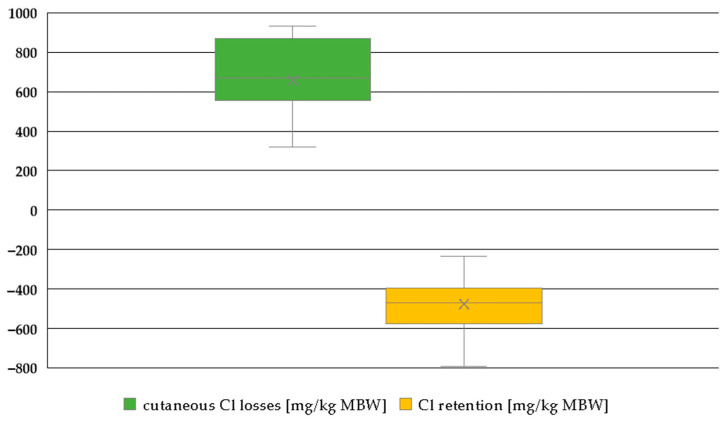
Box plot of Cl sweat losses and Cl retention (both in mg/kg MBW). The horizontal line represents the median, while the cross indicates the mean. Only data on working ponies, including information on Cl sweat losses, were available. Trials with water deficiency were excluded.

## Data Availability

Data presented in this study are available upon request from the corresponding author.

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
