# Peer review of "A Meta-Analysis on Quantitative Sodium, Potassium and Chloride Metabolism in Horses and Ponies"

_animals, 2025, doi:10.3390/ani15020191_

Round 1
Reviewer 1 Report
Comments and Suggestions for Authors
1. The study's introduction can be longer; you can include more background information related to the study's objectives.
2. Include more information about K & Cl metabolism in the Introduction.
3. Why did you include these particular studies in lines 81-83.
4. What do you mean by working animals and how did you quantify it.
5. Please explain in detail how you measured sweat losses in the study.
6. Please make sure to include all your results including the non-significant ones.
Reviewer 2 Report
Comments and Suggestions for Authors
The manuscript reviewed metabolism of sodium (Na), potassium (K), and chloride (Cl) regarding possible differences between ponies, and horses, the effect of exercise and the impact of water deficiency based on relevant literature. The review is thorough and informative. Importance of the said minerals in normal physiology has been mentioned.
Below are some comments in the spirit of helping the authors improve their manuscript.
Introduction: Introduction looks incomplete. The authors only talked about Na (negative effects due to deficiency or excess intake, daily requirement, digestibility etc.). Kindly discuss the other two minerals covered in the study.
Results: The authors mentioned ‘working animals’ in several places throughout the manuscript. Kindly mention its horse or pony.
Discussion: The discussion is thorough.
Kindly try to give some justification if available in literature for the differences in parameters (between ponies, and horses, between working and non-working animals and the impact of water deficiency etc).
Kindly add a paragraph on knowledge gap and potential future research area on the topic.
Reviewer 3 Report
Comments and Suggestions for Authors
This manuscript does an admirable job of evaluating studies that have examined intake of Na, K, and Cl, and accompanying digestibilities and retention. One concern is that retention is combined with sweat losses. I understand why that is done as most studies have not evaluated how much of these electrolytes are lost in the sweat – making it impossible to differentiate between the amount actually retained, and the amount that is lost in the sweat (and not retained). Again, this is understandable but it could be misleading as it would appear that there would actually be this amount retained in the body. There are numerous studies examining sweat rate and composition and it would be useful to discuss how retention might be influenced if sweat rate and composition were taken into consideration. Obviously huge amounts of electrolytes are lost in the sweat and are thus not retained. A major point behind this paper is evaluating differences between working and non-working horses. However, one of the major differences between these two classes of horses is working horses likely have huge sweat losses of electrolytes, while non-working horses would tend to have a relatively minor amount of sweat losses of electrolytes.
I would think that a paper evaluating electrolyte balance would tr y to calculate how these variables reported in this work would also try to (with calculations made from studies in where sweat amounts and composition were measured) estimate how these factors would be influenced if retention and sweat losses were separated. Again, it would only be an estimate, but it could present a much clearer message to readers if done.
Line 5. Both authors are probably not the department chair as signified by the superscript “1”.
Lines 73-76, the quotation marks need to be fixed.
Line 113. “This concerned to the data” is confusing. Please reword.
Line 124. I would suggest changing “after the studies completion” to “after the completion of the study” as the way it is currently worded it should be possessive.
Line 134. Change to “34,2” to “34.2”
Line 152. Change “0,05” to “0.05”
Figure 1. Ae there any data points above 100%? It doesn’t look like there are so, if that is the case, I would suggest just having the Y-axis go up to 100%.
Figure 3. Please consider changing the Y-axis to center around the Na serum concentrations that were found. In other words, instead of having it go from 0 to 180 mmol/l, possibly have it go from 120-160. That would allow readers to see more clearly where the data points were located.
Lines 208-213 (and other places). Some of these means are reported to a hundredth of a %. That is extremely precise – and probably incorrect. By reporting means to this degree of precision, it suggests that a degree of precision that is not likely realistic.
Figure 6. I would suggest having the upper % on the Y-axis to be 100 (as there are no data points above that).
Line 232. Reporting your P-value to the thousandths place provides no meaningful information when the P-value is 0.474. It may be useful when it is 0.004 or something like that – but reporting the P-value as either 0.5 or 0.47 is all you would need to do. This occurs in other areas of the manuscript (for instance, line 242) and should be corrected throughout.
Figure 8. It would be reasonable to have the Y-axis only go from 2 to 5 mmol/l instead of 0 to 6. This would allow the data to be viewed more clearly.
Figure 11. I would suggest having the Y-axis range between 40 and 100 (unless any of the intakes were above 100%, though it does not look like there were).
Figure 13. I would suggest the Y-axis to be between 90 and 130.
Lines 309-310 (and other places). The mean and the standard error should be reported to the same degree of precision. For instance, 61 and 4 instead of 61 and 3.9, and 93 and 4 instead of 93 and 4.1.
Figure 15. I might suggest the bottom range of the Y-axis start at -400 instead of -600.
Line 383. I would change “also explain, why” to “also explain why”.
Round 2
Reviewer 3 Report
Comments and Suggestions for Authors
After reviewing this version I am comfortable with the revisions and commend the authors for making the suggested revisions. In short, I would support this version being published.